# The Role of Phytohormones in Plant Response to Flooding

**DOI:** 10.3390/ijms23126383

**Published:** 2022-06-07

**Authors:** Xin Wang, Setsuko Komatsu

**Affiliations:** 1College of Agronomy and Biotechnology, China Agricultural University, Beijing 100193, China; 020350286@163.com; 2Faculty of Environmental and Information Sciences, Fukui University of Technology, Fukui 910-8505, Japan

**Keywords:** flooding, plants, rice, soybean, phytohormones, ethylene, abscisic acid

## Abstract

Climatic variations influence the morphological, physiological, biological, and biochemical states of plants. Plant responses to abiotic stress include biochemical adjustments, regulation of proteins, molecular mechanisms, and alteration of post-translational modifications, as well as signal transduction. Among the various abiotic stresses, flooding stress adversely affects the growth of plants, including various economically important crops. Biochemical and biological techniques, including proteomic techniques, provide a thorough understanding of the molecular mechanisms during flooding conditions. In particular, plants can cope with flooding conditions by embracing an orchestrated set of morphological adaptations and physiological adjustments that are regulated by an elaborate hormonal signaling network. With the help of these findings, the main objective is to identify plant responses to flooding and utilize that information for the development of flood-tolerant plants. This review provides an insight into the role of phytohormones in plant response mechanisms to flooding stress, as well as different mitigation strategies that can be successfully administered to improve plant growth during stress exposure. Ultimately, this review will expedite marker-assisted genetic enhancement studies in crops for developing high-yield lines or varieties with flood tolerance.

## 1. Introduction

Climatic conditions around the world are rapidly changing and causing imbalances in the environment [1]. Environmental variations are responsible for changes in precipitation patterns, which lead to negative effects on the development and production of crops [2]. Flooding is the most serious abiotic stress, as it induces growth retardation, yield reduction, and plant death [3]. Flooding causes complete or partial submergence stress, which has a deteriorative effect on seed germination, vegetative composition, and reproductive development of plants [4]. Flooding induces oxygen-deficient conditions, which adversely alter the growth stages of the plant life cycle [5]. Low oxygen conditions alter plant metabolism from aerobic respiration to anaerobic fermentation, and these metabolic changes are responsible for the impairment of plant development [6,7]. These findings clearly indicate that plant growth and reproductive development are seriously impaired by reduced oxygen during exposure to flooding stress.

On the other hand, plants have several strategies to survive submergence conditions. One strategy is quiescence, where plants do not elongate shoots under flooding conditions to minimize energy and carbohydrate consumption, and regrow when the flood recedes. This strategy is efficiently implemented by plants such as soybean and *Rumex acetosa*, both of which can survive relatively deep and transient floods [8,9,10,11,12]. Another strategy is escape, where plants rapidly extend their petioles or stems to allow leaves to reach the surface of the water in order to aerate the remainder of the plant. This strategy is utilized by plants such as deepwater rice and *Rumex palustris* [13,14,15,16,17]. These reports indicate that plants have developed several strategies to cope with the adverse consequences of flooding conditions.

Under flooding stress, plants differentially regulate proteins involved in hormone metabolism, signal transduction, transcriptional control, glucose degradation/sucrose accumulation, alcohol fermentation, gamma-aminobutyric acid shunt, suppression of reactive oxygen species (ROS) scavenging, mitochondrial impairment, ubiquitin/proteasome-mediated proteolysis, and cell-wall loosening [18,19,20]. In some of these events, plants balance the synthesis and transport of phytohormones and regulate the responses to waterlogging via complex signaling processes [21]. These reports indicate that plants can cope with flooding conditions by embracing an orchestrated set of physiological adjustments, which are regulated by an elaborate hormonal signaling network.

Phytohormones, such as ethylene, abscisic acid (ABA), and gibberellic acid (GA), orchestrate the acclimation response to flooding in rice [22]. Physical entrapment and biosynthesis of ethylene upon flooding have been proven to initiate the plant growth response. Low levels of ethylene induce *Sub1A-1* expression, which triggers the limitation of ethylene production during submergence [23]. In deepwater rice, flooding triggers degradation of ABA, which is an antagonist of GA, leading to enhancement of the GA response and the promotion of shoot elongation [24]. A decrease in ABA and an increase in GA biosynthesis were demonstrated in response to flooding that promoted the elongation of shoots necessary for leaf emergence from water [25]. Furthermore, ABA efficiently delayed ethylene-induced and GA-promoted cell death [26]. These results indicate that phytohormones such as ethylene, ABA, and GA act synergistically to confer plant adaptations to flooding conditions.

Phytohormones function as the central regulators for plant growth, development, and stress adaptation during flooding stress, and the signaling responses to phytohormones display crosstalk. During the past decades, the strategies employed by plants to adapt to or survive submergence have been revealed through quantitative trait locus mapping and omics approaches, highlighting the master roles of phytohormones. Herein, the morphophysiological changes of plants exposed to flooding are summarized, and plant adjustments to flood are revealed by phytohormone signaling induced by ethylene, ABA, GA, and others. Furthermore, the interplays of various phytohormone-mediated plant adaptation and survival strategies during flooding are discussed. This review focuses on phytohormonal response mechanisms of plants to flooding conditions which will open new opportunities for agricultural applications to improve plant growth under different flooding circumstances.

## 2. Morphophysiological Changes of Plants during Flooding

Flooding leads to either partial or complete submergence stress, and it imposes hazards for plant growth and development from seed germination to grain filling. Leaf petiole movement, stomatal closure, shoot elongation, and formation of adventitious roots/lateral roots/aerenchyma are obvious morphological alterations induced by hypoxia [13,27]. Carbon/nitrogen metabolism, photosynthesis, ion homeostasis, programmed cell death, hormone signaling, as well as other cellular metabolisms work in synergy in flooded plants [28,29]. Herein, the morphophysiological responses of plants to flooding are summarized (Table 1).

### 2.1. Arabidopsis

*Arabidopsis* attune morphological adaptation and physiological responses to flooding to maximize their efficacy or minimize stress impacts. The root is the first organ to sense flooding and the plants can replace the original root system with adventitious roots if the oxygen supply is scare [79]. Hypoxia inhibits growth of the main roots, while adventitious roots facilitate oxygen supply, resulting in improvement of root growth and plant survival. It was elucidated that auxin played an essential role in adventitious rooting by the hypocotyls of pre-etiolated flooded seedlings of Arabidopsis through activation of auxin transport by PIN-FORMED and AUXIN RESISTANT1/LIKE AUX1 [35]. The group VII ethylene response factor (ERFVII) *HRE2* promoted elongation of emerging adventitious roots [80], but inhibited hypoxia-induced root bending [31]. Additionally, a set of physiological responses are induced by hypoxia in *Arabidopsis* seedlings. For example, oxygen shortage resulted in potassium starvation in plant roots, and calcium-dependent signaling by *CIPK25* improved seedling capacity to maintain ion homeostasis [36]. ROS homeostasis, stomatal aperture, and chlorophyll degradation ultimately influence plant recovery from submergence through the activation of *SENESCENCEASSOCIATED GENE113*, *ORESARA1*, and *RESPIRATORY BURST OXIDASE HOMOLOG D* [32]. These studies show that interplay of hormone signaling with ROS homeostasis contributes to morphological alterations in response to flood and post-recovery flooding conditions.

### 2.2. Rice

Under flooding, deepwater rice accelerated underwater internode elongation via a boost in GA production with the aid of ethylene; however, semi-dwarf rice grown in paddy fields displayed short stems due to the low efficiency of GA biosynthesis, but sufficient GA was produced in the uppermost stem node to allow internode elongation in order to produce floral clusters above the leaf canopy during flowering [81,82]. In wild rice, *Oryza grandiglumis*, quiescence was not mediated by *Sub1A* as previously reported for deepwater rice, and the escape response based on internode elongation was induced by high moisture content instead of ethylene [38]. A comparative study between deepwater and non-deepwater rice showed that GA biosynthesis, trehalose biosynthesis, fermentation, cell wall modification, ethylene signaling, and jasmonic acid (JA) metabolism were significantly different between these two cultivars, and JA participated in internode elongation during the submergence of deepwater rice [42]. Aside from stem elongation, the formation of radial oxygen loss (ROL) in rice roots is an induced by waterlogging, and ABA-induced suberin lamellae formation in the exodermis facilitated ROL barrier formation [47,83]. Meanwhile, low concentrations of organic acids, specifically acetic, propionic, butyric, and hexanoic acid, triggered a ROL barrier in roots but with a slight effect on root extension [44]. Additionally, rice roots developed on the soil surface to protect plants from stresses in paddy fields and increase rice yields based on the regulation of *DEEPER ROOTING 1* involved in root system architecture [46]. These findings suggest that cross talk of GA, ethylene, and JA directs stem elongation, and ABA and organic acids meditate ROL formation in flooded rice.

### 2.3. Soybean

The influence of flooding stress on soybean morphological alterations are evidenced by the suppression on radicle protrusion, root structure, stem elongation, and photosynthesis pigment [84]. Oxygen deficiency suppressed nitrogen fixation by the nodules of soybeans due to the reduced activity of nitrogenase and downregulation of *asparagine synthetase* and *glutamic acid decarboxylase*; however, these effects were reversible after draining [53]. Flooding induced a reduction in net photosynthesis that could be relieved by the accumulation of starch granules and the application of exogenous ethephon [48,54]. A comparative study between a waterlogging tolerant soybean line and a sensitive line showed better development of adventitious roots/aerenchyma cells in stele and a higher ethylene production ratio contributed to flourishing plants during waterlogging [51]. In addition, a series of physiological responses was responsible for retarded growth in flooded soybeans, such as mitochondrial impairment, cell wall loosening, proteosome-mediated degradation, and sucrose accumulation [85]. However, enhancement of transcriptional regulation/protein homeostasis, suppression of cell death, and activation of auxin-triggered transcriptomic modifications for saving energy worked synergistically to help soybean survive short-term flooding [55,84]. These findings show that auxin and ethylene are involved in soybean adaptation to flood conditions through the regulation of adventitious root formation and energy conservation.

### 2.4. Wheat

In wheat, cell death in seminal roots and the short length of adventitious roots induced by hypoxia resulted in a low root:shoot ratio, and deficient nitrogen efficiency hampered shoot growth and grain yield [86]. An increase in adventitious root number and aerenchyma formation within these roots are characteristic of wheat tolerance to waterlogging, and nitric oxide was necessary to induce aerenchyma, either upstream of or in parallel with ethylene [56,86]. Aerenchyma formed 10 mm from the root tip and was mediated by ROS that initially accumulated in mid-cortex cells [59]. Waterlogging also induced ROS accelerated programmed cell death in endosperm cells when wheat seedlings were exposed to stress at the flowering stage [60]. Similarly, during seed germination, the degree of emulsification and endosperm cell degradation was increased and the number of amyloplasts was reduced in waterlogged wheat compared with the control [61]. Additionally, a comparative study was carried out in two wheat cultivars with contrasting submergence tolerance, and leaf gradation was faster and phytol/malondialdehyde content was higher in the sensitive cultivar compared the tolerant genotype, indicating that deprivation of ROS contributed to wheat tolerance to submergence [57]. These reports highlight the detrimental effects of flooding on seed germination and leaf senescence, but the acceleration of ROS scavenging might relieve these symptoms in stressed wheat.

### 2.5. Other Plants 

The effects of flooding stress on plant growth and development have also been observed in other plants. In barley, the average yield was reduced up to 50% by waterlogging [87]; however, sustained biomass, increased yield, and retention of chlorophyll were observed in barley with loss-of-function of *proteolysis6* [63]. Activation of depolarization-activated outward-rectifying potassium channels can improve potassium retention and confer hypoxia stress tolerance in barley [64]. In maize, waterlogging exerts severe impacts on the roots rather than the shoots [65], limiting root function due to hypoxia perturbed auxin flow/distribution, which is required to establish a quiescent center, and suppressing phytoglobins that can stabilize root performance under low oxygen conditions [66]. In addition, an improvement in root architecture and enhancement of chlorophyll biosynthesis through overexpression of *ZmCAO1* enhanced maize tolerance to waterlogging [67]. Moreover, adventitious root formation, carbohydrate metabolism, ROS accumulation, and hormone signaling were responsible for hypoxic conditions in tomato, cucumber, grapevine, and lotus [69,70,72,73,75], indicating that flooding stress can induce general alterations in plants.

## 3. Ethylene Signaling in Plants during Flooding

Flooding is detrimental for plants mainly due to restricted gas exchange underwater, which further induces rapid accumulation of ethylene in plant cells [88]. Ethylene is considered the major regulator of plant adaptation to flooding stress, such as raising plants by aerenchyma, facilitating gas diffusion by adventitious roots, directing antithetical strategies for plant survival by shoot elongation, and elevating leaves above water based on hyponastic growth [89]. Ethylene is also extensively characterized as an essential phytohormone for recovery after hypoxia by replenishing tricarboxylic acid cycle substrates [90,91], and post-submergence induced ethylene could interplay with ROS, photoinhibition, and desiccation to modulate plant growth during anoxia reoxygenation [32,92]. In this section, ethylene signaling in plant responses to flooding stress are summarized (Figure 1, Table 2).

### 3.1. Arabidopsis

Aquaporins help plants maintain water balance and gas transport under hypoxic conditions, and ethylene promotes water transport rate via phosphorylation of AtPIP2;1 [93]; in addition, the abundance of RAP2.12, an ERF-VII factor, is involved in the regulation of aquaporin activity under the control of *HCR1* [106]. *RAP2.12* negatively regulated root bending induced by hypoxia, showing that the growth direction of primary roots is antagonistically regulated by low oxygen and hypoxia-activated ERF-VIIs [31]. Overexpression of *HRE2*, a member of the ERF-VIIs, induces elongation of adventitious roots, suggesting that ERF-VII transcription factors enhance the establishment of adventitious roots under the negative control of ethylene [80]. Ethylene stimulates auxin synthesis in cotyledons, which was then translated to epidermal cell layers in the hypocotyl and interacted with GA and brassinosteroid (BR) to induce hypocotyl elongation [107]. Besides the roles of ethylene in root architecture and hypocotyl elongation, ethylene signaling was also responsible for leaf movement, as evidenced by the *ethylene-SHYG-ACO5* loop for the onset of rapid petiole cell expansion during waterlogging stress [30]. These results show that ethylene has versatile roles in root architecture, hypocotyl elongation, and leaf movement via the diverse functions of ERF-VIIs.

### 3.2. Rice

A plethora of findings have demonstrated the effects of ethylene on seed germination and plant growth in rice exposed to flooding. Rice seeds can germinate in absence of sufficient oxygen with lengthening of the coleoptile due to smart sugar management mediated by cross interaction of alpha-amylase activity, calcium signaling, and trehalose-6-P-phosphate content [108,109]. A reduction in the germination rate during submergence stress could be mitigated by blocking ethylene signaling; however, ethylene priming improved seedling survival during submergence through the activation of antioxidant responses mediated by hydrogen peroxide [43]. Ethylene effects on plant height have been addressed by studies on *ERFs* such as *Sub1A* [110], *SNORKEL1*/*2* [15], and *OsEIL1a* [41]. *Sub1A* dampens ethylene production and GA responses, leading to quiescent growth of seedlings during submergence [23], and *SNORKEL1*/*2* and *OsEIL1a* boosted stem elongation via enhancement of GA and conferred an escape strategy on deepwater rice [15,41]. However, not all rice varieties respond to submergence dependent on ethylene. For example, the wild rice *Oryza grandiglumis* displayed enhanced growth based on internode elongation during partial submergence of mature plants, but no significant ethylene accumulation was observed at the internode; similarly, the growth of shoots was not improved by complete submergence or exogenous ethylene owing to a lack of *Sub1A* [38,111]. These reports highlight the importance of ethylene signaling for rice adaptation to flooding and address how ethylene directs stem elongation.

### 3.3. Soybean

Compared with the excellent works in rice, the roles of ERFs in soybean have not been reported yet; however, a series of studies indicate that ethylene signaling participates in the soybean response to flooding. Soybean seeds are sensitive to sowing fields with uneven soil leveling, resulting in the poor performance of germination and seedling establishment [112,113]. Initial flooding during seed germination suppressed root growth, whereas application of ethylene promoted soybean growth through the enhancement of protein phosphorylation in root tips [76]. Similarly, exogenous ethylene relieved waterlogging induced symptoms via activated adventitious root initiation, increased root surface area, improved photosynthesis pigment, and an enhanced capacity for ROS scavenging [54]. The expressions of *GmXTHs*, which plays a role in cell wall extensibility through the modification of xyloglucan chains or catalysis of the hydrolysis of xyloglucan, were associated with ethylene and flooding, suggesting that ethylene signaling participated in flood-induced cell wall loosening [97]. A comparative study between flood-tolerant and -sensitive soybean showed that the genes related to ethylene biosynthesis were induced by the onset of stress and to higher levels in the roots of the tolerant genotype [98]. In another study, inconsistent changes in hormone profiles were observed based on the characteristics of soybean tolerance to waterlogging, and of note, a higher ratio of ethylene production was observed in the tolerant variety than the sensitive lines [51]. These reports indicate that a high level of ethylene accumulation correlates with improved tolerance to flooding stress through the modification of protein phosphorylation, ROS scavenging, and cell wall loosening.

### 3.4. Other Plants

Flood-induced ethylene signaling is responsible for root architecture, water transport, energy metabolism, and programmed cell death in maize, wheat, tomato, cucumber, lotus, and kiwifruit. Ethylene signaling responds to the onset of submergence stress with a sharp upregulation of the *ACC synthases* involved in ethylene biosynthesis [75]; meanwhile, increased ethylene levels stimulate auxin accumulation, which in turn, increases ethylene production to facilitate the formation of adventitious roots [72]. Ethylene regulation in flooded plants has been further proved with the identification of *ERFs*. For example, *ERF ZmEREB180* improved maize survival during waterlogging through prompt primordial root initiation and activated ROS scavenging [99]. Similarly, ectopic expression of *AdRAP2.3*, a member of *ERFs* in kiwifruit, enhanced tobacco tolerance to waterlogging with increased survival rate/plant biomass and prompt formation of aerial roots with the aid of increased activities of pyruvate decarboxylase and alcohol fermentation compared with wild type seedlings [102]. A study in petunia found that *PhERF2* could bind to the *ADH1-2* promoter and gain-of-function of *PhERF2* increased seedling tolerance to waterlogging, indicating that ethylene signaling participated in anaerobic fermentation to generate energy for proper functioning [103]. These reports show versatile roles of *ERFs* are involved in plant tolerance to flooding based on ROS scavenging, alcohol fermentation, and initiation of aerial roots.

## 4. Abscisic Acid Signaling in Plants during Flooding

It is widely accepted that plant responses to flooding stress are mainly regulated by ethylene signaling, and the interplays of ethylene and other phytohormones have been documented [27]. ABA signaling is associated with quiescent or escape strategies employed by plants based on the differential induction of ABA-dependent pathways, which can direct stem elongation though interactions with ethylene and GA [114]. Although the roles of ABA in flooding responses have been overshadowed by ethylene and GA, it participates in the emergence of adventitious roots, formation of secondary aerenchyma, hyponastic growth under hypoxia, as well as recovery from hypoxia [115,116]. ABA signaling in plant response to flood is presented below (Figure 2, Table 3).

### 4.1. Arabidopsis

Ethanolic fermentation is essential to mitigate the energy crisis induced by flooding [13]. The enzyme pyruvate decarboxylase plays pivotal roles in fermentation [128], and overexpression of kiwifruit *pyruvate decarboxylase1* in *Arabidopsis* conferred tolerance to waterlogging on seedlings, whereas the application of ABA caused a drop in the transcriptional level of *pyruvate decarboxylase* and resulted in inhibition of seed germination and root growth [117]. *AKIN10* is a conserved energy sensor, and its activity has an inverse modulatory role in plant adaptation to seawater through the regulation of *ABRE* and *RD22* promoter activities based on the protein stability of AtMYC2 [118]. Furthermore, ABA signaling has been associated with plant tolerance to post-recovery flood. For example, ABA signaling was a determinant for sensitivity to flooding in *abi2-1* mutant plants, which were ABA insensitive and presented better survival during submergence than wild-type adult seedlings, indicating that enhanced ABA sensitivity has a negative impact on post-submergence recovery [34]. In addition, expression of *SAG113*, a positive regulator of ethylene signaling, was elevated during flood recovery, but was decreased by the treatment of an ABA antagonist, suggesting interplays of ABA and ethylene on stomatal opening and dehydration [32]. These results show that ABA signaling appears to negatively regulate tolerance of Arabidopsis for flooding.

### 4.2. Rice

ABA signaling has been proved to contribute to oxygen transport, anaerobic germination, and stem elongation in flooded rice. A barrier to ROL enhances oxygen transport to the root apex and ABA signaling induced suberin lamellae formation in the exodermis; additionally, ABA application rescued the barrier that could not form in the mutant *osaba1* exposed to waterlogged soil [47]. It has been verified that miR393a contributed to retardation of coleoptile elongation and inhibition of stomatal development when seeds were submerged during germination. ABA application also promoted expression of *miR393a*, which was inhibited by submergence [39]. Furthermore, *OsVP1*, a transcription factor involved in ABA signaling, was a potential candidate for improving anaerobic germination tolerance in rice [119]. In addition, ABA signaling is responsible for suppression of stem elongation in rice during submergence. The accumulation of ethylene in flooded stems functions as an initial signal to enhance internodal elongation; it also reduces the amount of ABA and leads to an enhancement of GA-induced stem elongation, conferring an escape strategy on deepwater rice [129]. These reports show that increased ABA content inhibits seed germination and stem elongation during flooding by either working alongside or interacting with GA responses.

### 4.3. Soybean

Flooding stress reduces the accumulation of ABA in roots and leaves of soybean seedlings in spite of tolerance to flood, and a lower level of ABA was observed in flood-tolerant varieties than sensitive lines [48,55]. ABA signaling in flooded soybean appears controversial. Similar to the findings for Arabidopsis and rice, ABA is a negative player in conferring soybean flooding tolerance. For example, ABA inhibited the elongation of cells derived from phellogen during secondary aerenchyma formation in flooded soybean [120], which was consistent with the finding that downregulation of ABA contributed to well-developed aerenchyma cells [51]. On the other hand, application of ABA improved soybean flood tolerance in response to initial stress and during post-recovery stages. ABA content did not change in response to flooding during seedling establishment [22], but ABA altered the phosphorylation status of nuclear proteins in response to initial flooding stress [124]. In another study, application of ABA (10 μM) increased soybean tolerance to initial flood through activation of proteins involved in protein synthesis and RNA regulation [121]. Furthermore, exogenous ABA (10 μM) improved plant survival during flood through the modulation of energy conservation, enhancement of lignification, and inhibition of *cytochrome P450 77A1* [121,122,123]. Although the role of ABA signaling in soybean flood tolerance seems contentious, ABA (10 μM) application could be utilized to improve plant tolerance to short-term flooding.

### 4.4. Other Plants

Survival strategies employed by flooded plants have been studied in the wild dicot species *Rumex acetosa* and *Rumex palustris* with the finding that accumulated ethylene induced enhancement of ABA signaling and reduced GA in *R.*
*acetosa*, resulting in growth suppression and quiescence; in contrast, high amounts of ethylene downregulated ABA and upregulated GA to initiate the growth response in *R. palustris* [25,114,130]. ABA depletion induced by flooding stress has also been found in tomato [69,70], lotus [75], cucumber [125], *Solanum dulcamara* [126], and *Carrizo citrange* [127]. In tomato seedlings, 24h waterlogging caused a reduction in ABA content and induced expression of ABA receptors and ABA-dependent transcription factors; when stress was terminated, a reduction in gas exchange parameters was only observed in wild-type plants but not in the ABA-deficient genotype, suggesting that ABA depletion functions as a positive player in tomato responses to short-term flooding [70]. A comparative study was performed in cucumber seedlings with contrasting tolerance to waterlogging and the authors found that waterlogging exerted a dramatic drop in ABA in the hypocotyls of the tolerant variety compared with untreated seedlings due to accumulated ethylene; however, no significant ABA alterations were observed in the sensitive cultivar, resulting in suppressed growth of adventitious roots [125]. These results show that enhanced ABA signaling is a negative player in the formation of adventitious roots induced by flood.

## 5. Other Phytohormone Signaling in Plant during Flooding

A series of publications has proven that ethylene is the primary signal for plant adaptation to flooding; additionally, ethylene modulates a hormone cascade of ABA, GA, and auxin to induce adventitious rooting, internode elongation, and carbohydrate degradation [109,131]. Additionally, the effects of BR, JA, and salicylic acid (SA) on improving plant tolerance to flood are either independent or dependent on ethylene signaling to facilitate plant growth through activation on formation of adventitious roots, shoot elongation, photosynthetic pigment, and ROS scavenging [132,133,134,135]. Herein, phytohormone signaling, including GA, auxin, BR, JA, and SA, is described (Figure 3; Table 4).

GA is associated with plant responses to flooding, which is evident in the biosynthesis and signal transduction of GA in flooded plants [131]; additionally, exogenous GA relieves the hazardous effects of flooding on plant growth [50]. The role of GAs in plant adaptation to flooding has been well documented in rice and *Rumex* species, both of which are able to utilize quiescent and escape strategies to survive hypoxic conditions based on interplay with ethylene [9,15,23,110]. For the quiescent strategy, increased amounts of ethylene induced by flooding exert a reduction of GA along with an enhancement of ABA signaling, resulting in inhibition of plant growth in *Rumex acetosa* [9]. For the escape strategy, flooding increased ethylene, leading to an increase in GA to promote stem elongation; consequently, increased GA signals activate ethylene biosynthesis, resulting in positive feedback for ethylene and GA [131]. It has been observed that activation of *OsGA20ox2* facilitated the biosynthesis of GA and activated internode elongation in deepwater rice at the six-leaf stage [37]. Furthermore, *SD1*, which specifically binds *OsEIL1a*, was proven to increase GA accumulation in deepwater rice [41]. The antagonistic regulation of the GA response by stem elongation in rice was proven with two genes named *DEC1* and *ACE1*, which inhibited and boosted stem elongation, respectively, based on GA-mediated cell division independent of ethylene signaling [81]. These reports illuminate the essential roles of GA in mediating stem elongation in flooded plants; in particular, the outcomes obtained from deepwater rice should shed light on other crops.

Auxin has been implicated in promoting adventitious rooting by flooded plants. In wheat seedlings, upregulation of *TDC*, *YUC1*, and *PIN9* involved in auxin biosynthesis and transport contributed to high levels of auxin, which was required for nodal root induction during hypoxia [100]. In *Arabidopsis*, loss-of-function of *aux1* reduced the number of adventitious roots, whereas this symptom was rescued by exogenous auxin; additionally, enhancement of *AUX1*/*AFB2*/*PIN1* and suppression of *LAX1*/*LAX3*/*PIN4*/*PIN7* improved adventitious rooting by pre-etiolated flooded seedlings [35]. In rice, auxin receptors *OsTIR1* and *OsAFB2*, which are negatively regulated by miR393a, participated in elongation and stomatal development in coleoptiles during seed germination and seedling establishment during submergence [39]. Furthermore, interplays between auxin and ethylene have been identified that improve plant adaptation to flooding. For example, hypoxia reduced the abundance of auxin efflux carrier PIN2; however, overexpression of *RAR2.12* exacerbated this symptom, indicating the antagonistic control of auxin and ethylene [31]. In addition, ROS signaling was proven to mediate adventitious rooting inwaterlogged cucumber through the regulation of auxin and ethylene signaling [72]. These results elucidate interactions of the auxin response, ethylene signaling, and ROS homeostasis in root architecture and coleoptile growth in plants under hypoxic conditions.

An overall downregulation of genes related to auxin, BR, and GA biosynthesis was observed in the roots of waterlogged grapevine, indicating a general inhibition of root growth [73]. However, application of 24-epibrassinolide minimized the adverse impact of waterlogging on soybean through the improvement of root/leaf anatomy, an increase in chlorophyll accumulation, and enhancement of ROS scavenging [133]. BR biosynthesis was regulated in *Sub1A*-dependent manner and BR limited GA levels during shoot elongation in rice during submergence [132]. Additionally, JA and SA have also been verified as regulators of plant adaptation to flood via modulation of antioxidant metabolism and root architecture [134,135]. Gain-of-function of *RAP2-4* enhanced Arabidopsis seedling tolerance to waterlogging and its expression was induced by JA through interaction with the JA response element [138]. Similarly, JA synthesis in the meristematic region of roots was inducible by *ZmPgb1.2*, which positively regulated maize tolerance to anaerobic conditions [139]. Regarding SA enhancement of wheat tolerance to waterlogging, it was revealed that SA promoted formation of axile roots independently of ethylene, but its effect on adventitious rooting was ethylene-dependent [135]. These findings suggest that BR, JA, and SA appear to be positive players for plant flooding responses, though the transport and signaling pathways need further investigation to address their function in plant adaptation to flood.

## 6. Conclusions and Future Perspective

The elucidation summarized in this review suggests that phytohormones play major roles during plant response to flood and post-recovery submergence, ranging from signal transduction to cellular metabolism arrangements underlying seed germination, root architecture, stem elongation, petiole movement, and other processes (Figure 4). Furthermore, exogenous phytohormones have been proved to mitigate the effects of flooding on plants through the activation of flood tolerance capacity, suggesting that pre-treatment of phytohormones can be used to evoke flood-tolerance strategies for agricultural applications. Phytohormone signaling that mediates two contrasting responses to flooding has been well-elucidated in rice and *Remix* species based on the identification of several genes, such as *Sub1A*, *SNORKEL1/2*, *OsEIL1a*, *SD1*, *DEC1*, and *ACE1*, which direct the quiescence response or rapid internode elongation to help plant survival during submergence. Although fruitful achievements regarding hormone signaling in flooded rice could shed light on other plants, few genes contributing to flood tolerance have been identified due to the shortage of flood-tolerant materials and the complexity of tolerance characteristics, particularly for root structure. Currently, our understanding of the phytohormonal response to flood is mainly derived from ethylene, ABA, GA, and auxin, which could function alongside one another or in conjunction with other compounds. However, signaling transduction induced by BR, JA, and SA in flooded plants are still elusive because the receptors and transporters involved with these phytohormones remain ambiguous. With continuous advancements, multiple disciplines have boosted the development of omics, which have accelerated our understanding of plant responses to flooding stress. Therefore, new tools, including high-throughput interpretation of tolerant traits, single-cell based omics, and gene editing, will dissect the genes related to phytohormone signaling in flooded plants and assist in solving the complicated interplays among different phytohormones, facilitating the development of flood-tolerant plants based on the adaptive responses mediated by phytohormones.

## Figures and Tables

**Figure 1 ijms-23-06383-f001:**
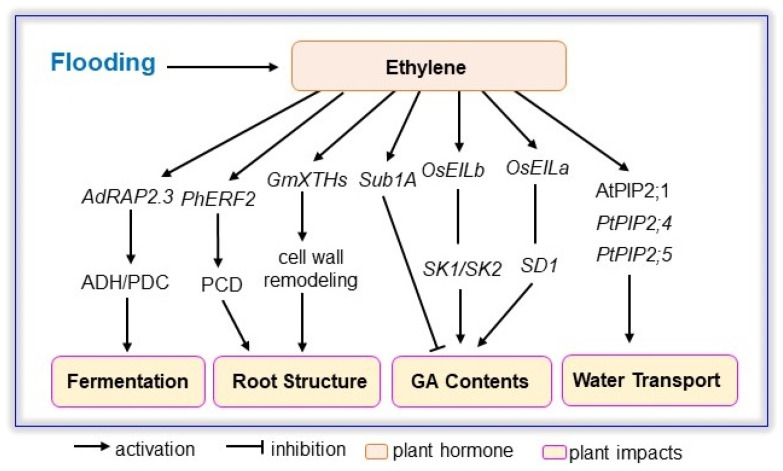
Ethylene signaling involved in plant response to flooding. Flooding increases ethylene amount and subsequently induces *RAP2.3*, *ERF2*, *Sub1A*, *EILa/b*, *PIP2;1/2;4/2;5* to modulate fermentation, root architecture, GA homeostasis, and water transport capacity in Arabidopsis, rice, soybean, *Actinidia deliciosa*, Petunia *(Petunia × hybrida)*, and *Populus tremuloides*. ADH, alcohol dehydrogenase; *ERF2, ethylene response factor2; EIL, ethylene insensitive like*; GA, gibberellic acid; PCD, programmed cell death; PDC, pyruvate decarboxylase; PIP, plasma membrane intrinsic protein; *RAP2.3,*
*related to apetala*
*2.3*; *SD1, SEMI-DWARF1; SK1/SK2*, *SNORKEL**1/SNORKEL2; Sub1A, Submergence1A; XTH, xyloglucan endotransglycosylases/hydrolases*.

**Figure 2 ijms-23-06383-f002:**
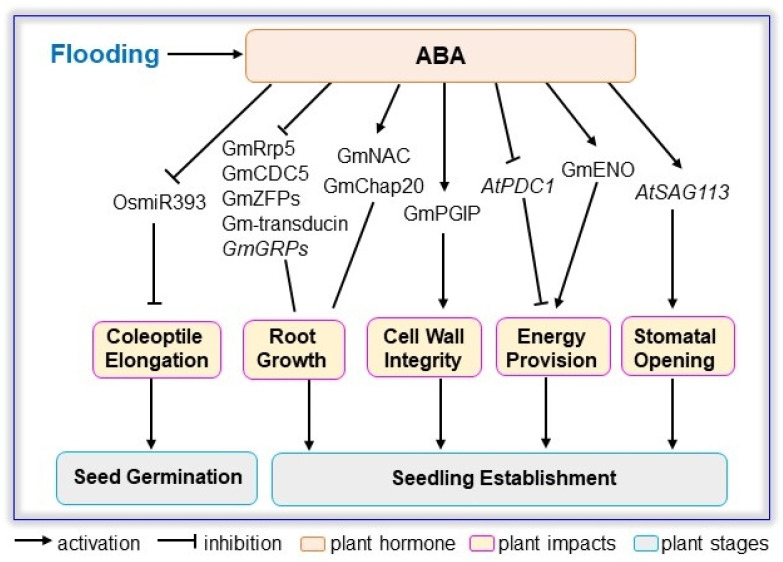
ABA signaling involved in plant response to flooding stress. Flooding alters ABA content with different accumulation patterns. The impact of exogenous ABA on flooded plants during seed germination and seedling establishment are summarized based on reported genes and proteins related to coleoptile elongation, root growth, cell wall integrity, energy provision, and stomatal opening in Arabidopsis, rice, and soybean. ABA, abscisic acid; CDC5, cell division cycle 5; Chap20, chaperone 20; ENO, enolase; *GRPs, glycine rich proteins*; NAC, nascent polypeptide associated complex; *PDC1, pyruvate decarboxylase1*; PGIP, polygalacturonase inhibiting protein; Rrp5, RNA binding rRNA processing protein 5; *SAG113, senescence-associated gene 113*; ZFPs, zinc finger proteins.

**Figure 3 ijms-23-06383-f003:**
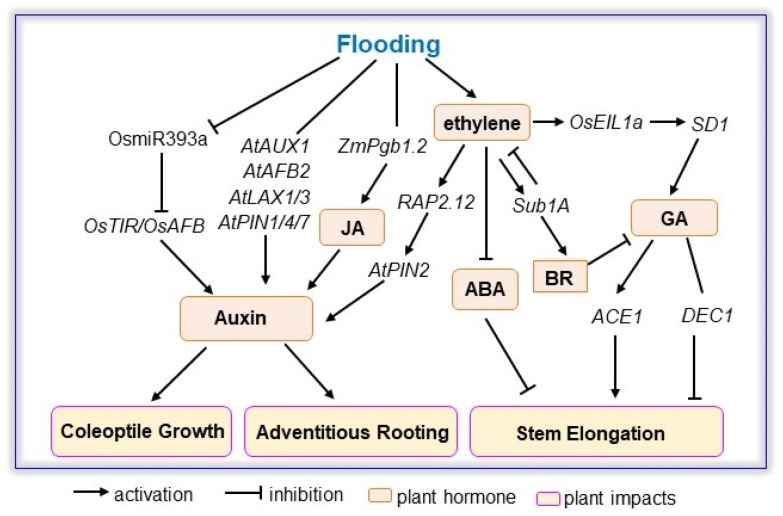
Interplays of GA, ABA, auxin, BR, ethylene, and JA signaling involved in plant response to flooding. The scheme of GA, ABA, auxin, BR, ethylene, and JA signaling associated with coleoptile growth, adventitious rooting, and stem elongation was constructed based on identified genes in Arabidopsis, rice, and maize. Flooding suppresses expression of miR393, which inhibits *OsTIR* and *OsAFB*, altering auxin mediated coleoptile growth during seed germination. Although auxin receptors and transports are affected differently by flooding, activation of *PIN1/AUX1/AFB2* and suppression of *LAX1/LAX3/PIN4/PIN7* improves adventitious rooting through auxin signaling. Meanwhile, gain-of-function of *RAP2.12* activates *AtPIN2* for auxin transport. The interaction between JA and auxin is evident in the enhancement of JA biosynthesis in plants overexpressing *ZmPgb1.2*. Flooding induces expression of *Sub1A* and rice bearing *Sub1A* presented high amounts of BR, leading to a reduction in GA content. Furthermore, flood-induced ethylene activates *OsEIL1a* that binds to *SD1*, leading to an increase in GA, which modulates stem elongation through *ACE1* and *DEC1.* ABA, abscisic acid; *ACE1, ACCELERATOR OF INTERNODE ELONGATION1*; *AFB, auxin signaling F-box; AUX, auxin resistant;* BR, brassinosteroid; *DEC1, DECELERATOR OF INTERNODE ELONGATION1; EIL1a, ethylene insensitive like1a;* GA, gibberellic acid; JA, jasmonic acid; *LAX, like AUX; Pgb, phytoglobin; PIN, plasma membrane intrinsic protein; RAP2.12*, *related to apetala 2.12; SD1, SEMI-DWARF1; Sub1A, Submergence1A; TIR, transport inhibitor resistant*.

**Figure 4 ijms-23-06383-f004:**
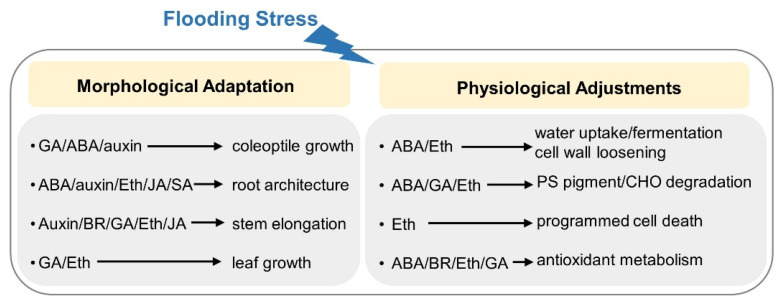
Summarizing overview of phytohormone-mediated morphophysiological alteration in flooded plants. Phytohormone-mediated morphological adaptation in flooded plants, including coleoptile growth, root architecture, stem elongation, and leaf petiole movement, is summarized. In addition, physiological adjustments induced by phytohormones under flooding are indicated, including water absorption, anaerobic respiration, cell wall loosening, photosynthetic pigment, carbohydrate degradation, programmed cell death, and antioxidant metabolism. ABA, abscisic acid; BR, brassinosteroid; CHO, carbohydrate; Eth, ethylene; GA, gibberellic acid; JA, jasmonic acid; SA, salicylic acid; PS, photosynthetic.

**Table 1 ijms-23-06383-t001:** Morphophysiological responses of plants to flood conditions.

Species	Morphophysiological Response	Ref.
*Arabidopsis*	Waterlogging induced hyponastic leaf growth mediated by ACO5 involved in ethylene biosynthesis.	[30]
	Hypoxia enhanced the angle of root bending by altering auxin signaling at root apex.	[31]
	Ability to maintain ROS homeostasis was required for post-flooding tolerance, and ethylene accelerated dehydration and senescence of plants during recovery.	[32]
	Waterlogging retarded seedling growth and a dense cuticle layer protected plants from diverse water conditions.	[33]
	Epigenetic regulation was associated with sensitization to flood from juvenility to adulthood.	[34]
	Proper phosphorylation of PINs and auxin transport direction were essential for AR formation in pre-etiolated flooding seedlings.	[35]
	Hypoxia induced potassium limitation and CIPK25 modified potassium flux in seedling roots.	[36]
Rice	GA sensitivity or transport was involved in leaf expansion and internode elongation in *O. sativa*.	[37]
	Partial submergence promoted internodal elongation of *O. grandiglumis* stem segments to the same extent as completely submerged *O. sativa* stem segments.	[38]
	Auxin signaling mediated by miR393a was involved in coleoptile elongation and stomatal development during seed germination and seedling establishment.	[39]
	Increased activity of amylase and decreased activities of ADH/PDC facilitated direct-seeded rice under waterlogging.	[40]
	Increased GA_4_, whose biosynthesis was transcriptionally mediated by ethylene-responsive factor, promotes internode elongation in deepwater rice to adapt periodic flooding.	[41]
	JA metabolism participated in submergence-mediated internode elongation in deepwater rice.	[42]
	Reduced chlorophyll content of leaves and induced leaf senescence in seedlings were relieved by ethylene precursor.	[43]
	Acetic, propionic, butyric, and hexanoic acids triggers inducible barrier to ROL in rice under waterlogging conditions.	[44]
	Stagnant flooding reduced grain yield and tiller number, but it increased plant weight and ethylene production.	[45]
	Shallower root growth angle enhanced rice yield in saline paddy field.	[46]
	ABA induced formation of suberin lamellae in exodermis contributed to ROL barrier formation in rice.	[47]
Soybean	Accumulation of starch granules and plastoglobules play roles in flooding tolerance.	[48]
	Amounts of lateral roots and total root mass influenced photon emission in flooded soybean.	[49]
	Reduced chlorophyll content was rescued by exogenous GAs via an increase in bioactive GA and enhancement of ABA biosynthesis.	[50]
	Well-developed aerenchyma cells in stele and AR contribute to flooding tolerance.	[51]
	ABA content, sugar levels, and circadian clock work to fine-tune photosynthesis and energy utilization in flooded plants.	[52]
	Flooding caused a decrease in asparagine and a concomitant accumulation of GABA.	[53]
	Flooding suppressed plant height, root structure area, chlorophyll content, chlorophyll fluorescence, and amino acid content; however, these were rescued by exogenous ethylene.	[54]
	ABA and ethylene coordinated transcriptomic energy-saving processes in response to flood.	[55]
Wheat	Hypoxia-induced NO was essential for aerenchyma development, and it accelerated ROS production, lipid peroxidation, and protein tyrosine nitration in wheat.	[56]
	Submergence induced faster leaf degradation and higher levels of phytol/malondialdehyde in sensitive cultivars than tolerant wheat.	[57]
	Waterlogging at tillering impaired photosynthetic activity in leaves and determined oxidative injury of roots.	[58]
	Well-developed aerenchyma was formed after 72 h of waterlogging, and it started in the mid-cortex cells and was regulated by ROS.	[59]
	Waterlogging increased ROS content in endosperm cells and eventually accelerated PCD.	[60]
	Waterlogging elevated the degree of emulsification and the degradation rate of endosperm cells; however, it reduced the number of amyloplasts in the endosperm.	[61]
	Waterlogging reduced internode lignin content and cytokinin mediated lignin deposition in plant biomass.	[62]
Barley	N-end rule pathway was a key regulator of the waterlogging response in barley, and reduced *HvPRT6* expression enhanced plant tolerance with sustained biomass, enhanced yields, and retention of chlorophyll.	[63]
	Hypoxia-induced potassium loss in roots was correlated with cell ability.	[64]
Maize	Compared with shoots, the roots were more impaired by waterlogging as evidenced by reduced length, area, and biomass.	[65]
	Hypoxia limited root function by perturbing auxin flow and distribution for establishment of an oxidized redox in the quiescent center, but these impacts were alleviated by overexpression of phytoglobin.	[66]
	Enhancement of chlorophyll biosynthesis by *ZmCAO1* confers maize tolerance to waterlogging.	[67]
	Lateral roots were able to form a barrier to ROL as sites for oxygen loss under hypoxic conditions.	[68]
Tomato	IAA allowed enhancement of root hydraulic conductivity in mycorrhizal plants under hypoxic conditions.	[69]
	ABA depletion improved water absorption during long-term waterlogging and maintained the relative water content in leaves.	[70]
	Reduced alcohol fermentation and enhanced glycolysis induced by hypoxia stress priming was crucial for flood tolerance.	[71]
Cucumber	Levels of ethylene, auxin, and ROS accumulated in waterlogged plants; ROS mediated ethylene- and auxin-induced AR formation.	[72]
Grapevine	Increased accumulation of GABA, succinic acid, and alanine at initial hypoxia remained at high levels within one week after recovery.	[73]
Lotus	Waterlogging reduced the number of stem/leaves, dry biomass of shoot/root, and chlorophyll fluorescence in tetraploid lotus.	[74]
	Water-repellent leaves became greasy, petioles of newly developed leaves became slim and long within the initial submergence, and almost all leaves fell off after stress was expended.	[75]
Barrel clover	Waterlogging decreased accumulation of raffinose, sucrose, hexoses, and pentoses in roots, and it increased starch, sugar, and phenolics content in leaves.	[76]
*Melilotus siculus*	Petiole reorientation towards vertical under partial and full submergence; however, constant rate of petiole extension and accumulation of sugar were only present during partial submergence.	[77]
*Taxodium ascendens*	Soil water table with middle level promoted formation of knee roots via induced ethylene and IAA production.	[78]

Ref, reference; ABA, abscisic acid; ADH, alcohol dehydrogenase; AOC5, ACC oxidase 5; AR, adventitious root; CIPK25, calcineurin β-like interacting protein kinase 25; CAO, chlorophyll A oxygenase; GA, gibberellic acid; GABA, gamma-aminobutyric acid; IAA, indole acetic acid; JA, jasmonic acid; PCD, programmed cell death; PDC, pyruvate decarboxylase; PIN, PIN-FORMED; ROL, radial oxygen loss; ROS, reactive oxygen species.

**Table 2 ijms-23-06383-t002:** Ethylene signaling in plants during flooding stress.

Plant Species	Ethylene Signaling during Flood	Ref.
*Arabidopsis*	Ethylene increased water transport rate in leaf cells by enhancing S280/S283 phosphorylation at the C terminus of PIP2;1.	[93]
	Ethylene inhibited formation of adventitious roots and overrode *HRE2*-induced adventitious root elongation.	[31]
	Hypoxia induced the primary roots to grow sidewise, and it was inhibited by *RAP2.12*.	[80]
	Leaf movement was reduced by root flooding in seedlings with loss-of-function of *aco5*.	[30]
Rice	Accumulation of ethylene did not induce internode elongation in *Oryza grandiglumis.*	[38]
	Ethylene signaling played a negative role in seed germination, but improved seedling tolerance to submergence through regulation of the antioxidant response.	[43]
	Stagnant flooding accelerated ethylene production and *sub1*-introgression into rice cultivar increased plant sensitivity to stagnant flooding.	[45]
	*OsEIL1a* specifically bound to *SD1* promoter, resulting in increased GA content, and allowed deepwater rice tolerance to submergence.	[41]
	*Sub1a* governed distinct responses to submergence in mature and growing leaves through regulation of ethylene, ABA, JA, and auxin.	[94]
	*Sub1a* promoted precocious photoautotrophy and restricted underwater elongation during seed germination.	[95]
Soybean	Exogenous ethylene promoted soybean growth during initial flooding through regulation of protein phosphorylation.	[96]
	Exogenous ethylene mitigated waterlogging stress in soybean with increased root surface, improved photosynthesis, and higher accumulation of protein/glutathione S-transferase.	[54]
	Ethylene was associated with cell wall remodeling in flooded soybean through regulation of expression of *GmXTHs*.	[97]
	Ratio of ethylene production was higher in a waterlogging tolerant line than sensitive soybean.	[51]
	Expression of genes related to ethylene biosynthesis were enhanced in roots of flood-tolerant soybean.	[98]
Maize	*ZmEREB180* positively regulated maize waterlogging tolerance through regulation of adventitious root formation and ROS homeostasis.	[99]
Wheat	Waterlogging effects on root growth, emergence and elongation of axile and lateral roots, and aerenchyma formation were associated with upregulation of *ACS7* and *ACO2* involved in ethylene biosynthesis.	[100]
Tomato	Flooding stress induced *ERF.E1*, *E2*, *E3,* and *E4* in tomato seedlings.	[101]
Cucumber	Accumulation of ethylene induces formation of adventitious roots during waterlogging and stimulated accumulation of auxin, which in turn increased ethylene content.	[72]
Lotus	Expression levels of *NnACSs* involved in ethylene biosynthesis were significantly upregulated in lotus during initial submergence.	[75]
Kiwifruit	Overexpression of *AdRAP2.3* enhanced tobacco tolerance to waterlogging via enhanced activities of PDC and ADH.	[102]
Petunia	*PhERF2* bound to *ADH1-2* promoter and positively contributed to waterlogging tolerance through modulation of programmed cell death and fermentation.	[103]
*Chrysanthemum morifolium*	High levels of ethylene accumulated in waterlogging tolerant variety compared with sensitive lines during waterlogging and reoxygenation conditions.	[104]
*Populus tremuloides*	Exogenous ethylene improved water transport via increased expression levels of *PIP2;4* and *PIP2;5* in roots of trembling aspen under hypoxic conditions.	[105]
*Taxodium ascendens*	Ethylene was induced by the middle water table, resulting in an enhancement of flood tolerance through the improvement of root ventilation.	[78]

Ref, reference; ABA, abscisic acid; ACO, 1-aminocyclopropane-1-carboxylic acid oxidase; ACS, 1-aminocyclopropane-1-carboxylic acid synthase; ADH, alcohol dehydrogenase; EIL, ethylene insensitive like; ERF, ethylene response factor; GA, gibberellic acid; HRE, hypoxia responsive ERF; JA, jasmonic acid; RAP, related to apetala; PDC, pyruvate decarboxylase; PIP, plasma membrane intrinsic protein; ROS, reactive oxygen species; SD, SEMI-DWARF; Sub1, Submergence1; XTH, xyloglucan endotransglycosylases/hydrolases.

**Table 3 ijms-23-06383-t003:** ABA signaling in plants to flooding stress.

Plant Species	ABA Signaling during Flood	Ref.
*Arabidopsis*	ABA interacted with ethylene to control stomata opening, dehydration, and senescence through *SAG113* and *ORE1* during submergence recovery.	[32]
	Application of ABA (10 μM) negatively regulated the expression of *pyruvate decarboxylase1* in response to waterlogging stress.	[117]
	Hypoxia induced *AKIN10* and controlled protein stability of AtMYC2, which was activated by ABA signaling in response to seawater flooding.	[118]
	ABA signaling was a determinant for plant sensitivity to submergence and impaired ABA signaling due to loss-of-function of *abi2-1* enhanced plant survival during submergence.	[34]
Rice	Inhibition of ABA biosynthesis suppressed the formation of a ROL barrier and development of suberin lamellae under stagnant conditions.	[47]
	ABA suppressed expression of *miR393a*, which regulated coleoptile elongation for seed germination during submergence.	[39]
	ABA content fluctuated at a low level during seed germination when submerged, which was mediated by *OsVP1.*	[119]
Soybean	ABA inhibited elongation of cells derived from phellogen during secondary aerenchyma formation induced by flooding.	[120]
	Flooding reduced ABA content in leaves of plants during flooding, especially the tolerant cultivar exposed to flooding for 15 days.	[48]
	Downregulation of ABA in response to flooding contributed to well-developed aerenchyma cells.	[51]
	Diurnal expression of *TOC1* was found in soybean exposed to flooding, and higher expression of *TOC1* coincided with elevated ABA content.	[52]
	Application of ABA (100 μM) could not protect plants from 14 days of waterlogging.	[54]
	Submergence reduced the abundance of bioactive ABA in roots and leaves in seedlings, and a low amount of ABA was sufficient to trigger ABA effects involved in quiescence.	[55]
	Application of ABA (10 μM) improved soybean tolerance during initial and survival stages of flooding through regulation of energy conservation, enhanced cell wall integrity, and inhibition of cytochrome P450 77A1.	[22,121,122,123]
	ABA signaling affected soybean responses to initial flood through phosphorylation of BTB domain containing protein 47, glycine rich protein, and rRNA processing protein Rrp5 in root tips.	[124]
Tomato	ABA depletion positively regulated plant tolerance to short-term waterlogging via improved gas exchange, activated NO metabolism, and ERF-VII stability.	[70]
	ABA was decreased in the roots of flooded mycorrhizal and non-mycorrhizal plants, whereas no significant difference was observed in leaves.	[69]
Cucumber	In a waterlogging-sensitive cultivar, ABA content in waterlogged hypocotyls decreased approximately 63.9% compared with the control, but no significant difference was observed in the tolerant variety.	[125]
Lotus	Decrease in ABA was implied by downregulation of *NcNCEDs* and upregulation of *NnCYP707A*, which was related to ABA biosynthesis and degradation, respectively, in response to initial submergence.	[75]
*Rumex acetosa*	Enhanced ABA signaling facilitated growth suppression in *Rumex acetosa* through a combination of maintained ABA levels and elevated receptor levels.	[9]
*Solanum dulcamara*	Drop in ABA was examined in stem and adventitious root primordia tissue in flooded plants, whereas ABA application (1 mM) prevented activation of adventitious root formation induced by flooding.	[126]
*Carrizo citrange*	ABA depletion constituted a specific response to flooding, and the conjugation involved in ABA metabolism was complementary to degradation in order to maintain ABA homeostasis.	[127]

Ref, Reference; ABA, abscisic acid; ABI, ABA insensitive; ERF, ethylene response factor; NCED, 9-cis epoxycarotenoid dioxygenase; OsVP1, Oryza sativa Viviparous1; ROL, radial oxygen loss; SAG, senescence-associated gene; TOC1, Timing of CAB expression1.

**Table 4 ijms-23-06383-t004:** Effects of GA, auxin, BR, JA, and SA in plants during flooding stress.

Hormones	Plant Species	Hormone Signaling	Ref.
GA	Rice	Flooding increased GA_1_ and GA_4_ content due to activation of *OsGA20ox2*, and GA induced internode elongation during the six-leaf stage in deepwater rice.	[37]
		*SD1* largely increased the content of GA_4_ via interaction with *OsEILa* and promoted internode elongation in deepwater rice.	[41]
		GA promoted elongation of the internode based on the reduction of JA content in deepwater rice.	[42]
		*DEC1* and *ACE1* inhibited and accelerated stem elongation, respectively, through GA-mediated cell division in the internode.	[81]
	*Rumex acetosa*/*R. palustris*	GA signaling was activated by initial submergence of both of *Rumex acetosa* and *R. palustris*, whereas sensitization to GA prior to its accumulation was induced in *R. palustris*.	[9]
	Barley	*Sln1d.8* allele, which encodes DELLA that can be degraded by GAs, improved seedling tolerance to flooding independent of root porosity.	[136]
	Soybean	Application of GA_4+7_ (50 μM) ameliorated the effects of short-term flooding through enhancement of glutathione activity, chlorophyll content, and NO signaling.	[50]
	Lotus	Slight increase in GA within the first 24 h of submergence was partly due to upregulation of *NnGA3OX1/2/3/4*.	[75]
Auxin	*Arabidopsis*	Hypoxia reduced protein abundance of PIN2, and overexpression of *RAR2.12* rescued this symptom.	[31]
		Activation of *PIN1/AUX1/AFB2* and suppression of *LAX1/LAX3/PIN4/PIN7* improved the development of adventitious roots.	[35]
	Rice	Submergence inhibited expression of miR393a, which negatively regulated auxin receptors *OsTIR1* and *OsAFB2* to mediate elongation and stomatal development in coleoptiles.	[39]
	Maize	*ZmPgb1.1* sustained PIN-mediated auxin and oxidized environment in the quiescent center of root apical meristems of seedlings during hypoxia.	[66]
	Cucumber	Waterlogging increased auxin-enhanced ethylene biosynthesis and they synergistically promoted formation of adventitious roots.	[72]
	*Solanum dulcamara*	Compared with partial submergence, complete submergence impaired the growth of adventitious root due to disruption of auxin instead of ABA and JA.	[137]
BR	Rice	During submergence, rice bearing *Sub1A* had a high content of BRs, which resulted in lower amounts of GA to restrict shoot elongation.	[132]
	Soybean	24-epibrassinolide (10 nM) relieved effects of waterlogging on soybean with improved root anatomy, photosynthetic pigment, ROS scavenging, and increased biomass.	[133]
	Grapevine	A general downregulation of genes related to BR, auxin, and GA biosynthesis was induced by waterlogging along with inhibition of root growth and lateral expansion.	[73]
JA	*Arabidopsis*	Rapid accumulation of JA during post-recovery of submergence interacted with antioxidant pathways to enhance seedling tolerance through JA-activated *MYC2*.	[134]
		Overexpression of *RAP2-4* isolated from *Mentha arvensis* enhanced *Arabidopsis* tolerance to waterlogging, and its expression was induced by JA via interaction with the JA response element.	[138]
	Maize	Overexpression of *ZmPgb1.2* conferred plant waterlogging tolerance and induced genes related to JA biosynthesis in the meristematic region of roots.	[139]
SA	Wheat	Application of SA (1 mM) improved wheat tolerance to waterlogging and its promotion of axile root formation and aerenchyma was ethylene independent and dependent, respectively.	[135]

Ref, Reference; ACE1, ACCELERATOR OF INTERNODE ELONGATION1; AFB, auxin signaling F-box; AUX, auxin resistant; BR, brassinosteroid; DEC1, DECELERATOR OF INTERNODE ELONGATION1; EIL, ethylene insensitive like; GA, gibberellic acid; JA, jasmonic acid; LAX, like AUX; Pgb, phytoglobin; PIN, plasma membrane intrinsic protein; ROS, reactive oxygen species; SD1, SEMI-DWARF1; Sub1, Submergence1; TIR, transport inhibitor resistant.

## Data Availability

Not applicable.

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
