# Peer review of "The Role of Phytohormones in Plant Response to Flooding"

_ijms, 2022, doi:10.3390/ijms23126383_

Round 1

Reviewer 1 Report

This review focused on phytohormonal response mechanisms of plants toward flooding stress, in addition to the employment of various mitigation strategies that can be successfully administered to improve plant growth on stress exposure. The manuscript is well structured and well discussed. However, some points should be checked and corrected before its acceptance in this journal. 

Therefore, according to my comments, I recommended the publication of the paper after major revision.

  • Please add and improve the phytohormonal response mechanisms of plants with good additional figures.
  • The study's background should be clearly stated. Describe the introduction and review of the work (Please add more information).
  • In Conclusion, the authors should add the significance of this research and its potential practical application.
  • The MS English needs to be improved. The article's English must be carefully checked for grammatical errors.

Author Response

Reviewer 1

This review focused on phytohormonal response mechanisms of plants toward flooding stress, in addition to the employment of various mitigation strategies that can be successfully administered to improve plant growth on stress exposure. The manuscript is well structured and well discussed. However, some points should be checked and corrected before its acceptance in this journal. Therefore, according to my comments, I recommended the publication of the paper after major revision.

Please add and improve the phytohormonal response mechanisms of plants with good additional figures.

Answer: Thank you very much for the comment. The phytohormonal response mechanisms of plants have been prepared with new Figure 4 in section “6. Conclusion and Future Prospective”. The correction has been marked with red color.

The study's background should be clearly stated. Describe the introduction and review of the work (Please add more information).

Answer: We are sorry for the unclear information. As required, background of this manuscript has been stated in the last paragraph of section “1. Introduction”. The description has been marked with red color in revised manuscript.

In Conclusion, the authors should add the significance of this research and its potential practical application.

Answer: Thank you very much for the suggestion. As commented, the significance of this research and its practical application have been written in section “6. Conclusion and Future Prospective” with red color in revised manuscript.

The MS English needs to be improved. The article's English must be carefully checked for grammatical errors.

Answer: We are sorry for these mistakes. The English writing of this manuscript has been improved and the grammatical errors have been corrected in revised manuscript. Additionally, this article has been corrected by native speaker of English.

Reviewer 2 Report

The manuscript ijms-1716583 addresses The role of Phytohormones in Plant Response to Flooding and is titled as such. It is intended as a detailed literature review article and is structured quite well. All components of the text are correct. The tables showing the response of each seed plant genus/species are valuable and convincingly described. I have no complaints about the Figures prepared, along with their description presented. Hormonal signaling is also discussed and synthetically shown in a tabular overview, a rather rare but valuable procedure. Overall, the authors demonstrated a good knowledge of the issues described. I wonder, however, if they should not put some issues in a more concise way, which would be beneficial for readers less familiar with the discussed topics.

Author Response

Reviewer 2

The manuscript ijms-1716583 addresses the role of Phytohormones in Plant Response to Flooding and is titled as such. It is intended as a detailed literature review article and is structured quite well. All components of the text are correct. The tables showing the response of each seed plant genus/species are valuable and convincingly described. I have no complaints about the Figures prepared, along with their description presented. Hormonal signaling is also discussed and synthetically shown in a tabular overview, a rather rare but valuable procedure. Overall, the authors demonstrated a good knowledge of the issues described.

I wonder, however, if they should not put some issues in a more concise way, which would be beneficial for readers less familiar with the discussed topics.

Answer: Thank you very much for the comment. To improve the readability of this manuscript, we have prepared new Figure 4 to summarize the phytohormonal response mechanisms of plants in section “6. Conclusion and Future Perspectives”. In addition, descriptions of this manuscript’s background and discussed topics have been summarized in section “1. Introduction”. These corrections have been marked with red color in revised manuscript.

Round 2

Reviewer 1 Report

Requested corrections were completed.